# Gaze-Contingent Eye-Tracking Training in Brain Disorders: A Systematic Review

**DOI:** 10.3390/brainsci12070931

**Published:** 2022-07-16

**Authors:** Laura Carelli, Federica Solca, Sofia Tagini, Silvia Torre, Federico Verde, Nicola Ticozzi, Roberta Ferrucci, Gabriella Pravettoni, Edoardo Nicolò Aiello, Vincenzo Silani, Barbara Poletti

**Affiliations:** 1Department of Neurology and Laboratory of Neuroscience, Istituto Auxologico Italiano, I.R.C.C.S., 20149 Milan, Italy; federica.solca@gmail.com (F.S.); silviatorre.psy@gmail.com (S.T.); f.verde@auxologico.it (F.V.); n.ticozzi@auxologico.it (N.T.); e.aiello@auxologico.it (E.N.A.); vincenzo@silani.com (V.S.); b.poletti@auxologico.it (B.P.); 2“Rita Levi Montalcini” Department of Neurosciences, University of Turin, 10126 Turin, Italy; s.tagini@auxologico.it; 3Istituto Auxologico Italiano, I.R.C.C.S., U.O. di Neurologia e Neuroriabilitazione, Ospedale San Giuseppe, 28824 Piancavallo, Italy; 4Department of Pathophysiology and Transplantation, Dino Ferrari Center, University of Milan, 20122 Milan, Italy; 5Department of Health Sciences, Aldo Ravelli Center for Neurotechnology and Experimental Brain Therapeutics, International Medical School, University of Milan, 20122 Milan, Italy; roberta.ferrucci@unimi.it; 6Neurology Clinic III, ASST Santi Paolo e Carlo, 20142 Milan, Italy; 7Istituto di Ricovero e Cura a Carattere Scientifico (IRCCS) Ca’ Granda Foundation Maggiore Policlinico Hospital, 20162 Milan, Italy; 8Department of Oncology and Hemato-Oncology, University of Milan, 20122 Milan, Italy; gabriella.pravettoni@unimi.it; 9European Institute of Oncology, IRCCS, 20141 Milan, Italy; 10PhD Program in Neuroscience, School of Medicine and Surgery, University of Milano-Bicocca, 20126 Monza, Italy

**Keywords:** eye-tracking, gaze-contingent training, brain disorders, attention, inhibition

## Abstract

Eye movement abnormalities in association with cognitive and emotional deficits have been described in neurological, neurodevelopmental, and psychiatric disorders. Eye-Tracking (ET) techniques could therefore enhance cognitive interventions by contingently providing feedback to patients. Since no consensus has been reached thus far on this approach, this study aimed at systematically reviewing the current evidence. This review was performed and reported according to PRISMA guidelines. Records were searched for in PubMed, Web of Science, and Scopus (1990–2021) through the following string: (‘Eye Tracking’ OR ‘Eye-Tracking’ OR ‘Oculomotor’) AND (‘Neuropsychol*’ OR ‘Cognitive’) AND (‘Rehabilitation’ OR ‘Training’ OR ‘Stimulation’). Study outcomes were thematically classified and qualitatively synthesized. A structured quality assessment was performed. A total of 24 articles were included, addressing neurodevelopmental (preterm infants and children with autism spectrum disorder, Rett syndrome, or ADHD; *N* = 14), psychiatric (mood and anxiety disorders or alcohol dependence; *N* = 7), and neurological conditions (stroke; *N* = 3). Overall, ET gaze-contingent training proved to be effective in improving cognitive and emotional alterations. However, population heterogeneity limits the generalizability of results. ET gaze-contingent protocols allow researchers to directly and dynamically train attentional functions; together with the recruitment of implicit, “bottom-up” strategies, these protocols are promising and possibly integrable with traditional cognitive approaches.

## 1. Introduction

Eye-Tracking (ET) techniques are widespread Alternative and Augmentative Communication (AAC) systems that have been largely employed as a mean to administer neuropsychological tests in the absence of motor–verbal integrity in neurological patients [1,2,3]. Within such an approach, eye movements represent the only viable medium in order to visually select and provide responses when a task is administered. Moreover, eye movement alterations have been described as a valid marker for cognitive dysfunctions in neurological populations [4] For instance, antisaccade/prosaccade and smooth pursuit paradigms proved to be sensitive to the early detection of cognitive, frontal-like alterations in neurodegenerative conditions [5] such as amyotrophic lateral sclerosis [6,7,8], Parkinson’s disease [9,10], and Huntington’s disease [11,12].

Moreover, altered affective experiences, such as those observed in psychiatric (e.g., mood and psychotic disorders) [13,14,15,16] and neurodevelopmental conditions (e.g., autism spectrum disorders (ASD) and attention deficit hyperactivity disorder (ADHD)) [17,18], have been shown to be related to abnormal eye movement patterns, concerning both implicit aspects (i.e., smooth pursuit and saccadic eye movements) and more explicit, cognitive components (e.g., exploratory patterns and visual search). In psychiatric and neurodevelopmental disorders, attentional biases regarding emotional information have also been observed, e.g., in major depression [19], social anxiety [20,21], and ASD [22].

Thus, eye behaviors provide valuable information on both higher cognitive functions and emotional states in neurological, psychiatric, and neurodevelopmental populations. More specifically, not only saccades and smooth pursuit, but also fixation, pupil size, and eye blinks, represent salient features reflecting the efficiency of such processes [23].

Beyond providing a robust measure of exploratory biases and cognitive impairments, a less explored application of ET is its potential as a tool to be combined with treatments that target cognitive and emotional abilities in order to enhance the interventional outcomes in patients with different brain disorders. To the aim of cognitive interventions, gaze-contingent ET techniques [24] have been proposed as particularly useful [25], as they are able to deliver real-time, online feedback to users based on their eye movements, in contrast to approaches merely based on passively scanning looking patterns. However, little is known about the potential of gaze-contingent, ET-based cognitive interventions in the abovementioned neurodevelopmental, psychiatric, and neurological populations, to which such an approach is likely to be applicable—as in patients presenting with eye movement abnormalities—and which thus may benefit from it.

Given the above premises, the present review aims to focus uniquely on the use of ET as a mean to support cognitive interventions in heterogeneous clinical populations, in order to summarize available evidence on such an approach and provide an up-to-date, critical appraisal addressing its advantages and limitations.

## 2. Materials and Methods

This review was performed and reported according to the Preferred Reporting Items for Systematic Reviews and Meta-Analyses (PRISMA) guidelines [26]. This systematic review has not been pre-registered.

### 2.1. Eligibility Criteria

Randomized controlled trials, clinical trials, protocols for feasibility studies, and single case studies investigating the use of ET systems for cognitive training in any brain and psychiatric disorder were included. Participants of any age and clinical conditions were considered. Only studies adopting gaze-contingent ET protocols for training were considered, with eye movements used to trigger events on the screen in response to the participant’s gaze. Conversely, studies were excluded if ET techniques had been addressed only as a tool to either measure treatment outcomes or provide feedback signals in a closed-loop system (e.g., in vestibular treatment). Moreover, ET-based protocols developed for the training of sport, work, or medical fields (not for final clinical purposes) were excluded.

Only published manuscripts (articles, book chapters, and conference proceedings if published as full-text articles in peer-reviewed journals) written in English were addressed.

### 2.2. Information Sources

Studies were identified by searching electronic databases and scanning the article reference lists in November 2021. The search was applied to the electronic databases of PubMed (1990–2021), Web of Science (1990–2021), and Scopus (1990–2021).

### 2.3. Search Strategy

Studies were identified using a combination of the following terms: (‘Eye Tracking’ OR ‘Eye-Tracking’ OR ‘Oculomotor’) AND (‘Neuropsychol*’ OR ‘Cognitive’) AND (‘Rehabilitation’ OR ‘Training’ OR ‘Stimulation’). See Appendix A, for specific text.

### 2.4. Study Selection

Titles and abstracts were screened for relevance. A full-text analysis was then performed on the selected articles. Eligibility assessment was performed independently by two reviewers (LC, FS); disagreements between reviewers were resolved by consensus.

### 2.5. Data Collection Process

A data extraction sheet to summarize relevant results from the selected studies was developed, pilot-tested on five randomly-selected included studies, and refined accordingly. One review author (LC) extracted the data from the included studies, and the second author (FS) checked the extracted data. The final extraction form included the following outcomes: sample type and size, study designs, main quantitative and/or qualitative outcomes related to the review topics, Level of Evidence (LoE) classification (see next paragraph), and type of ET device adopted.

A post-hoc thematic classification of studies was performed according to the clinical condition addressed. Disagreements were resolved by discussion between the two review authors who performed the categorization; if no agreement could be reached, a third author decision (ST) was forecast. Ethics committee authorization was not required, as this study reviewed previously published data.

### 2.6. Quality Assessment and Risk of Bias

The quality evaluation of studies and outcomes included in the review focused on the following aspects: the LOE for the methodological quality of the study design (based on the 7-level rating scheme by Ackley and colleagues [27]); the sample size and representativeness; the heterogeneity of included studies within the identified thematic categories. The outcomes of this evaluation are summarized in the Results section and in Appendix A, as well as commented on in the Discussion section.

## 3. Results

The study selection process is depicted in Figure 1. A total of 24 papers met the eligibility criteria and were thus included. Their characteristics are summarized in Appendix A.

Twelve articles presented case-control studies addressing either clinical or symptomatic populations [28,29,30,31,32,33,34,35,36,37,38,39] or individuals being at risk for the development of clinical conditions (very preterm infants [40]; infants with a first-degree relative with a diagnosis of ADHD [41]); eight articles involved only healthy subjects [42,43,44,45,46,47,48].

Two records presented the protocol for subsequent feasibility studies [49] or clinical trials [50]; one is a usability study without a control group [43]; two are feasibility studies including only preliminary results [34,35].

Sample sizes were relatively small in the majority of the studies: only five studies included more than fifty subjects [25,36,39,46,48]; ten had 25–50 subjects [28,29,32,34,37,41,43,44,45,47]; and seven had 10–25 subjects [30,31,33,35,38,40,42].

All included studies were published in the last 11 years. Only three studies were published in 2015 or before. The least recent paper was published in 2011.

Reviewed studies were described according to the clinical population addressed. In particular, the following clinical conditions were identified: psychiatric (major depressive disorder, dysphoria, social anxiety disorder, or alcohol dependence), neurodevelopmental (preterm infants, children with ASD, Rett syndrome (RS), or ADHD) and neurological conditions (post-stroke cognitive impairment) (Appendix A).

The collected findings for each topic are described in detail in the following paragraphs.

### 3.1. Neurodevelopmental Conditions

The majority of the included studies are described in this section (14/24). In particular, ET-based training programs were developed to address the following populations: typically developing infants [44,45]; infants at familial risk for ADHD [41] or with an overt diagnosis of ADHD [30,31,37,42]; children with RS [34,35,39]; children with ASD [25,36]; and very preterm infants [40,49].

Besides the involvement of different populations, the included studies describe a variety of training programs targeting slightly heterogeneous abilities. For example, some protocols addressed selective/focused attention and interference-resolution, search for a changing target and ignoring distractors, visuo-spatial working memory, task-switching, visual search, and inhibition [41,44,45]. These authors’ protocol encompasses four gaze-contingent tasks, presented in the form of animals and cartoon characters, with different difficulty levels adaptively adjusting to patients’ performance.

The Attention Control Training proposed by Perra and colleagues [40,49] employed three types of tasks, presented by means of interactive cartoons, in order to train the following abilities: search for a target among distractors; visual short-term memory for objects embedded in scenes; and maintaining a goal.

Lee and colleagues [30,31] developed a six-module training program to improve different inhibitory abilities; in particular, sustained attention and impulse control, with three different difficulty levels.

Garcia-Baos and colleagues [37] compared an ET-based protocol to a mouse version of a computer game intended to train attention in ADHD, featuring continuous and contingent feedback and adaptive difficulty levels related to patients’ performances.

Caprì and colleagues [39] developed an interactive ET-based digital game for improving attention and motivation abilities in children with RS.

All the described programs proved to be effective in improving the target cognitive abilities in both passive [31,40,41,49] and active control groups [37]. In particular, the study by Garcia-Baos and colleagues [37] supported the idea that the gaze-contingent element of the training program is a crucial aspect for its effectiveness on attentional systems.

A different approach was proposed by Wang and colleagues [25,36], which focuses on training looking patterns of patients with ASD, based on normative data collected in healthy individuals, in order to promote more efficient gaze strategies and improvement in daily social interactions. The results suggested that gaze-contingent training effectively mitigates the decreases of attention towards faces in ASD, with larger training benefits being gained by children with a greater degree of cognitive impairment.

Other training protocols have been developed for educational purposes in schools, in the context of the COVID-19 pandemic and consequent e-learning and remote teaching programs [34,35]. In particular, children with multiple disabilities, such as those affected by RS, hardly benefit from remote education and training programs; the adoption of eye-based gaze interactions revealed promising preliminary results that were reliable, with respect to the standard Tobii ET [35], and effective in reducing stereotypies and increasing attention [34].

For similar learning purposes, Garcia-Zapirain and colleagues [42] developed an arithmetical game-based platform that allows participants to interact by means of eye movements and gestures. The results on the usability and acceptability of such a protocol appeared to be promising.

### 3.2. Psychiatric Conditions

Seven studies are included within this section. Five studies addressed depressive disorders (major depressive disorder [28]; dysphoric or depressive symptoms [29,46,48]; and “negative” mood [47]), one study addressed social anxiety disorders [32], and another alcohol dependence [33]. Three studies addressed non-clinical samples [46,47,48], while one other study consisted of a protocol description for a placebo-controlled trial which would involve a sub-clinical population of students identified as at-risk for depressive disorders based on a below- vs. above-cut-off score on the Beck Depression Inventory [29]. The remaining studies consisted of clinical trials on in- and outpatients [28,32,33].

All these investigations adopted the so-called “attentional bias modification” (ABM) treatment, which targets the dysfunctional attention patterns towards negative or threatening stimuli, featuring patients with depression and anxiety, respectively.

In such studies, the training encompassed emotional materials with positive/negative values, consisting of pictures [46], faces [28,29,47], or sentences [48]. The training addressed, as outcomes, both the disengagement from negative stimuli and the maintenance of attentional focus on positive ones.

The pre- and post-assessments included the evaluation of training effects directly on associated visual attention tasks, aimed at detecting attentional biases, as well as, at times, the evaluation of transfer effects to other critical, both cognitive and emotional, aspects, such as mood and depressive symptoms, emotional learning, motivation, and quality of life [28,29,46,47]. In selected patients with major depressive disorders, no significant improvements were reported, although a distal transfer of training to emotional memory was detected [28]. In unselected populations, the training led to an increase in mood reactivity and recovery after a stressor [47], and in reappraisal ability, together with an improvement in rumination ratings [48]. In all these studies, a direct effect on attentional bias was observed.

The study on alcohol dependence employed an ad-hoc training protocol, based on additional monetary feedback, contingent on the eye-gaze avoidance of alcohol-related stimuli [33], compared to an active control group where monetary reward was unrelated to gaze direction. The authors found that the program was effective with regard to trained images, while no transfer effects were observed with regard to craving behaviors and addiction symptoms, neither at the end of the intervention nor at the follow-up.

Finally, the study on social anxiety disorder [32] compared a gaze-contingent, music-based reward training, aimed at diverting attention towards neutral over threatening faces, to a passive control condition. Music was selected by the patients before the training started, in order to enhance and personalize the reward effect. The training led to a reduction in social anxiety symptoms, which was stable at the 3-month follow-up; moreover, the reduction in time spent looking at the threatening, trained faces was extended to untrained stimuli.

### 3.3. Neurological Conditions

Three studies addressing stroke patients are included within this section [38,43,50]. One of them presented a new protocol to develop cognitive rehabilitation tasks based on daily-life activities, without providing any preliminary results [50]. The study by Lévy-Bencheton and colleagues [38] aimed to improve homonymous visual field defects following stroke by means of a modified anti-saccade paradigm based on a bottom-up strategy. In such a task, the feedback target was systematically presented with an offset, with respect to the final eye position, in the direction of increasing eccentricity (outward) and with an amount equal to 10% of the actual size of the eye displacement. The comparison against such an approach proved to be, in a sub-group of patients, more effective in reading, visual exploration tasks, and visual quality of life, when compared to traditional, top-down strategies (i.e., voluntary saccade training).

The study by Verghese and colleagues [43] investigated the possible transfer of training of inhibitory control from oculomotor (antisaccade training) to manual response modality (assessed by means of the Simon Task [51]); moreover, the authors investigated if such a transfer generalized to other inhibitory tasks, as well as to non-inhibitory measures. The authors addressed, as the active control groups, those trained with fixation and prosaccade tasks, and showed a transfer of benefits only with regard to the antisaccade group that was selective for the Simon Task.

### 3.4. Risk of Bias

With reference to the study design, the majority of the included records had a LOE of II (randomized controlled studies). Only three studies had a LOE of IV (descriptive, qualitative study) [34,35,42]. The LOE classification was not applicable to four studies, which described protocols for randomized controlled trials [29,41,49] or for unspecified clinical trials [50] (Level of Evidence (LOE) classification: Level I—Evidence from a systematic review or meta-analysis of all relevant RCTs (randomized controlled trial), evidence-based clinical practice guidelines based on systematic reviews of RCTs, or three or more RCTs of good quality that have similar results; Level II—Evidence obtained from at least one well-designed RCT; Level III—Evidence obtained from well-designed controlled trials without randomization (i.e., quasi-experimental); Level IV—Evidence from well-designed case-control or cohort studies; Level V—Evidence from systematic reviews of descriptive and qualitative studies (meta-synthesis); Level VI—Evidence from a single descriptive or qualitative study; Level VII—Evidence from the opinion of authorities and/or reports of expert committees).

Despite the majority of the included studies consisted of randomized controlled trials, sample sizes were overall small, ranging from 12 to 86 participants. An a-priori sample size estimation was only seldom delivered [28,29,31,40,41,43,49].

According to a thematic perspective, the abovementioned categorization of the reviewed studies highlights an important bias related to the very small number of studies falling into the “neurological condition” group.

Within the applications of ET to psychiatric disorders, anxiety and alcohol dependence were under-represented, with only one study for each clinical population; similarly, among neurodevelopmental disorders, only one study addressed ASD patients [36].

Besides these specific clusters of patients, overall, sparse and heterogeneous findings were observed.

## 4. Discussion

The present review summarizes findings concerning ET-based, gaze-contingent cognitive and emotional training in populations with brain and psychiatric disorders. The majority of the studies here with reported focused on neurodevelopmental conditions (preterm infants, children with ASD, RS, ADHD), followed by psychiatric (major depressive disorder, dysphoria, social anxiety disorder, alcohol dependence) and neurological disorders (i.e., stroke).

Overall, applications in neurodevelopmental populations were aimed at improving attentional and inhibitory abilities [30,31,40,41,44,45,49], training looking patterns as compared to normative data [25,36], or promoting motivation and participation to e-learning activities in students with multiple disabilities [34,35,42]. Such protocols proved to be effective and well-accepted, even if against almost passive [31,40,41,49] or no control groups [25,34,35,42].

As to psychiatric conditions, ET trainings adopted the ABM paradigm, especially in respect to patients with depressive symptoms. Overall, promising findings were detected regarding both the modification of attentional biases and the transfer of treatment effects to other cognitive and emotional components, although such results appear to be biased, as they refer to selected sub-sample of patients with depression. Therefore, further evidence is needed in unselected and larger samples to promote the external validity of these findings [28,29,46,47,48]. With regard to social anxiety disorder, the training seemed effective in improving the core symptoms, despite needing further empirical support from future investigations in this population [32].

At variance with studies on psychiatric conditions, those addressing neurological diseases were sparser and more limited in number. However, the two clinical studies for which empirical evidence was documented reported promising and novel applications of the anti-saccade paradigm [38,43]. More specifically, the innovative aspects of such protocols are represented by both the general purpose—i.e., this paradigm is employed to adaptively train cognitive functions, and not to assess them—and specific targets and outcome measures addressed—i.e., visual field defects and possible transfer of the inhibitory training from the oculomotor to the manual response modality, respectively.

A number of the ET-based trainings herewith described for attentional deficits or biases represent interesting novel alternatives to less recent treatments, which come with several limitations. Standard approaches entail, as to neurodevelopmental and neurological disorders, computer-based cognitive trainings with gamification features [52]. As to psychiatric populations, programs based on the dot-probe paradigm are addressed to modify emotional attentional biases, by taking into account reaction times (RTs) [53].

However, such traditional, computer-based gaming approaches may not be feasible in infants or patients with neurological disorders affecting the motor system, due to possible limitations in motor abilities that are, by contrast, an unavoidable requirement to interact with the device. Moreover, in the same target samples, language skills needed to understand instructions might either be under-developed (infants) or impaired due to acquired brain lesions (neurological patients). Conversely, gaze-contingent ET addresses participant eye movements to trigger events on the screen, allowing them to interact and receive feedback by solely using their eyes. Interestingly, only one of the included articles actually compared such an ET-based gaze-contingent training with a more traditional approach, either computer- or non-computer-based [37]. It can be hypothesized that such comparisons are still lacking in the literature due to the fact that little or no consensus has been reached as to how an ET-based cognitive intervention program should be developed and standardized [54]. The absence of these investigations could also contribute the fact that several ET metrics have been shown to be promising ways to measure cognitive and emotional processes [23]; although this is, overall, an advantage, such a broad range of metrics might make it hard for researchers devoted to the field of ET-based cognitive interventions to orient themselves.

With regard to the psychiatric field, ET-based measures might provide better therapeutic targets compared with RT-based ABM protocols, as ET provides a continuous measurement of overt attention, at variance with key-press behavior that only captures indirect and less dynamic indexes of the underlying attentive processes. Moreover, RT-based attention bias modification protocols utilize monotonous trials, which are experienced by some patients as tedious, with a negative impact on their motivation and engagement.

Even though the reported ET-based approaches seem to overcome the described limitations, the presence of possible ET calibration difficulties, together with the costs of ET technology, restricts the adoption of such an approach in clinic- and home-based treatments.

The main limitations of the present review concern the above described clinical heterogeneity of the recruited samples, which limits the generalizability of the discussed results. From a review perspective, this systematic review summarized qualitative results, while quantitative data analysis was not performed. This choice was made in accordance with the presence of different approaches in the outcome measures and eye movement features described, which led to difficulty in summarizing and comparing such aspects across studies. Future research efforts should be spent in providing more detailed data about a limited range of outcome parameters (ET metrics, cognitive measures, etc.), at least within each specific clinical field.

In summary, ET gaze-contingent training proved to be effective in improving cognitive and emotional alterations in neurological, neurodevelopmental, and psychiatric conditions. The ET gaze-contingent protocols allow training directly and dynamically attentional functions, together with the recruitment of implicit, bottom-up strategies. The integration of such an approach with more traditional cognitive interventions could provide standard rehabilitation and psychological care with some additional, relevant, and effective features.

## Figures and Tables

**Figure 1 brainsci-12-00931-f001:**
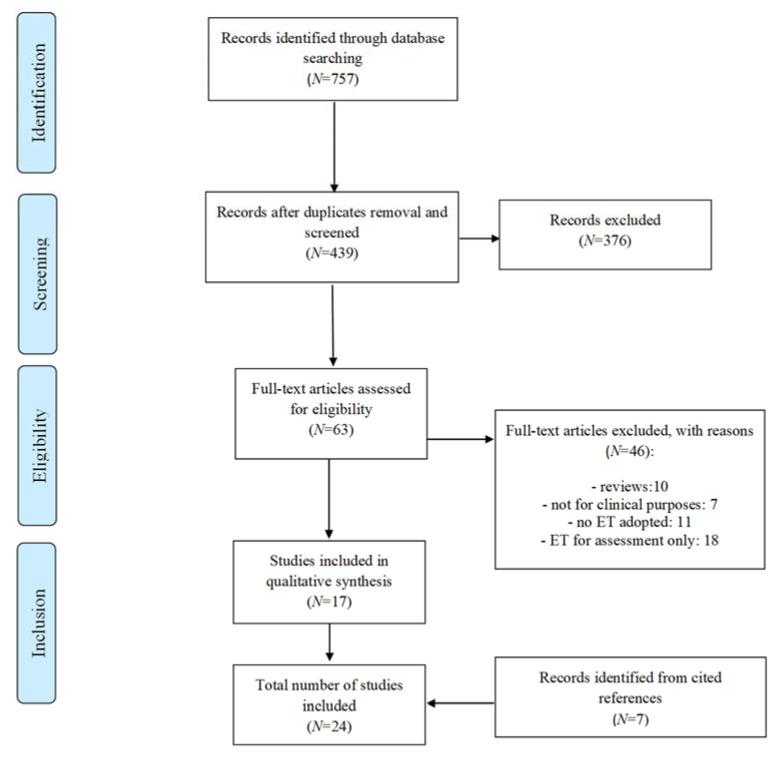
PRISMA flow-chart displaying study selection process. Diagram adapted from Moher et al., (2009).

## Data Availability

Data are contained within the article or Appendix A. The data presented in this study are available in Appendix A.

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
