# Peer review of "Gaze-Contingent Eye-Tracking Training in Brain Disorders: A Systematic Review"

_brainsci, 2022, doi:10.3390/brainsci12070931_

Round 1
Reviewer 1 Report
I appreciated having the opportunity to review this manuscript. This study conducted systematic review on ET techniques. They found that ET gaze contingent training proved to be effective in improving cognitive and emotional alterations. I have some comments and suggestions:
1. Is there any specific reasons why set the search start date as January 1990? If so, please include it in the manuscript.
2. The "Level of Evidence (LOE)" in line 118 on page 3 has been introduced in line 109 on the same page.
3. The information in Figure 1 is clear. It is better to explain why only 24 studies left in this systematic review? Does it because limited published works in this area? Or other reasons. Is there any other literatures mentioned it?
4. I think there is a typo in "The least recent paper was published in 2011". Do you mean "2021"?
5. The discussion on limitations of this systematic review is not sufficient. It is better to include limitation from both clinical prospective and review prospective. For example, this systematic review only conducted the qualitative analysis, but no quantitative data analysis. I am wondering if there is a chance to abstract data from most of the 24 articles and conduct quantitative analysis. If so, that will provide stronger evidence to address research question and investigate potential biases.
Author Response
1.Is there any specific reasons why set the search start date as January 1990? If so, please include it in the manuscript
Thank Reviewer 1 for your read and suggestions. With regard to the point 1, there is not a specific reason for having set the search start date at January 1990. We deleted the month from the date since it seemed not so useful for us, according to your question.
2.The "Level of Evidence (LOE)" in line 118 on page 3 has been introduced in line 109 on the same page.
We corrected this formal error in the manuscript.
3.The information in Figure 1 is clear. It is better to explain why only 24 studies left in this systematic review? Does it because limited published works in this area? Or other reasons. Is there any other literatures mentioned it?
Most of the literature that was found with key search terms concerned not clinical applications of eye movement assessment and training, in particular in the field of medicine, education, aviation and sport. Moreover, our selection was aimed at including only studies where gaze-contingent paradigm were used, i.e. with eye movement on-line control of interaction with the stimuli provided. For all such reasons, only 24 studies were finally included in the review.
4.I think there is a typo in "The least recent paper was published in 2011". Do you mean "2021"?
This was correct, since the oldest (least recent) study has been published in 2011 and not 2021.
5. The discussion on limitations of this systematic review is not sufficient. It is better to include limitation from both clinical prospective and review prospective. For example, this systematic review only conducted the qualitative analysis, but no quantitative data analysis. I am wondering if there is a chance to abstract data from most of the 24 articles and conduct quantitative analysis. If so, that will provide stronger evidence to address research question and investigate potential biases.
Thanks, Reviewer 1. We added limitations from a review perspective, as suggested, and we also explained in this section the reason why we decided to not conduct quantitative data analysis, i.e. due to the broad range of outcome measures described across the studies.
Reviewer 2 Report
The paper summarizes the research on the application of eye-tracking technology for studying brain disorders.
A presented literature review focused on clinical conditions and the number of subjects engaged in the experiments. However, there is a lack of a review regarding the analysed eye movement features and results obtained based on them. A recommendation of ET-based experiments for application in a given clinical condition should also be provided.
Author Response
The paper summarizes the research on the application of eye-tracking technology for studying brain disorders.
A presented literature review focused on clinical conditions and the number of subjects engaged in the experiments. However, there is a lack of a review regarding the analysed eye movement features and results obtained based on them. A recommendation of ET-based experiments for application in a given clinical condition should also be provided.
This systematic review summarized qualitative results, while less attention has been payed to quantitative data across the manuscript. This choice has been taken in accordance to the presence of different approaches in outcome measures and eye movement features in the reviewed studies, that lead to difficulty in summarizing and comparing such aspects. However, we now added a column in Supplementary Table 2 reporting the type of ET device employed in each study, in order to improve the review with some technical information, useful to capture the brioad range of device (and outcomes) involved.
Round 2
Reviewer 2 Report
Dear Authors,
Thank you for considering my suggestion.